# Understanding Translationese in Cross-Lingual Summarization

**Jiaan Wang**♠,*, **Fandong Meng**◇, **Yunlong Liang**♡, **Tingyi Zhang**♠
**Jiarong Xu**♣, **Zhixu Li**♠†**and Jie Zhou**◇

♠Shanghai Key Laboratory of Data Science, School of Computer Science, Fudan University, Shanghai, China

◇Pattern Recognition Center, WeChat AI, Tencent Inc, China

♡Beijing Key Lab of Traffic Data Analysis and Mining, Beijing Jiaotong University, Beijing, China

♣School of Management, Fudan University, Shanghai, China

{jawang.nlp,yunlonliang,zhangtingyi712}@gmail.com

{fandongmeng,withtomzhou}@tencent.com  {jiarongxu,zhixuli}@fudan.edu.cn

## Abstract

Given a document in a source language, cross-lingual summarization (CLS) aims at generating a concise summary in a different target language. Unlike monolingual summarization (MS), naturally occurring source-language documents paired with target-language summaries are rare. To collect large-scale CLS data, existing datasets typically involve translation in their creation. However, the translated text is distinguished from the text originally written in that language, *i.e.*, translationese. In this paper, we first confirm that different approaches of constructing CLS datasets will lead to different degrees of translationese. Then we systematically investigate how translationese affects CLS model evaluation and performance when it appears in source documents or target summaries. In detail, we find that (1) the translationese in documents or summaries of test sets might lead to the discrepancy between human judgment and automatic evaluation; (2) the translationese in training sets would harm model performance in real-world applications; (3) though machine-translated documents involve translationese, they are very useful for building CLS systems on low-resource languages under specific training strategies. Lastly, we give suggestions for future CLS research including dataset and model developments. We hope that our work could let researchers notice the phenomenon of translationese in CLS and take it into account in the future.

## 1 Introduction

Cross-lingual summarization (CLS) aims to generate a summary in a target language from a given document in a different source language. Under the globalization background, this task helps people grasp the gist of foreign documents efficiently, and attracts wide research attention from the computational linguistics community (Leuski et al., 2003;

Wan et al., 2010; Yao et al., 2015; Zhu et al., 2019; Ouyang et al., 2019; Ladhak et al., 2020; Perez-Beltrachini and Lapata, 2021; Liang et al., 2022).

As pointed by previous literature (Ladhak et al., 2020; Perez-Beltrachini and Lapata, 2021; Wang et al., 2022b), one of the key challenges lies in CLS is data scarcity. In detail, naturally occurring documents in a source language paired with the corresponding summaries in a target language are rare (Perez-Beltrachini and Lapata, 2021), making it difficult to collect large-scale and high-quality CLS datasets. For example, it is costly and labor-intensive to employ bilingual annotators to create target-language summaries for the given source-language documents (Chen et al., 2022). Generally, to alleviate data scarcity while controlling costs, the source documents or the target summaries in existing large-scale CLS datasets (Zhu et al., 2019; Ladhak et al., 2020; Perez-Beltrachini and Lapata, 2021; Bai et al., 2021; Wang et al., 2022a; Feng et al., 2022) are (automatically or manually) translated from other languages rather than the text originally written in that language (Section 2.1).

Distinguished from the text originally written in one language, translated text[1] in the same language might involve artifacts which refer to "translationese" (Gellerstam, 1986). These artifacts include the usage of simpler, more standardized and more explicit words and grammars (Baker et al., 1993; Scarpa, 2006) as well as the lexical and word order choices that are influenced by the source language (Gellerstam, 1996; Toury, 2012). It has been observed that the translationese in data can mislead model training as its special stylistic is away from native usage (Selinker, 1972; Volansky et al., 2013; Bizzoni et al., 2020; Yu et al., 2022). Nevertheless, translationese is neglected by previous CLS work, leading to unknown impacts and potential risks.

Grounding the truth that current large-scale CLS

---

*Work was done when Jiaan Wang was interning at Pattern Recognition Center, WeChat AI, Tencent Inc, China.

†Corresponding author.

[1] "Translated text" is equal to "translations", and we alternatively use these two terms in this paper.

datasets are typically collected via human or machine translation, in this paper, we investigate the effects of translationese when the translations appear in target summaries (Section 3) or source documents (Section 4), respectively. We first confirm that the different translation methods (*i.e.*, human translation or machine translation) will lead to different degrees of translationese. In detail, for CLS datasets whose source documents (or target summaries) are human-translated texts, we collect their corresponding machine-translated documents (or summaries). The collected documents (or summaries) contain the same semantics as the original ones, but suffer from different translation methods. Then, we utilize automatic metrics from various aspects to quantify translationese, and show the different degrees of translationese between the original and collected documents (or summaries).

Second, we investigate how translationese affects CLS model evaluation and performance. To this end, we train and evaluate CLS models with the original and the collected data, respectively, and analyze the model performances via both automatic and human evaluation. We find that (1) the translationese in documents or summaries of test sets might lead to the discrepancy between human judgment and automatic evaluation (*i.e.* ROUGE and BERTScore). Thus, the test sets of CLS datasets should carefully control their translationese, and avoid directly adopting machine-translated documents or summaries. (2) The translationese in training sets would harm model performance in real-world applications where the translationese should be avoided. For example, a CLS model trained with machine-translated documents or summaries shows limited ability to generate informative and fluent summaries. (3) Though it is sub-optimal to train a CLS model only using machine-translated documents as source documents, they are very useful for building CLS systems on low-resource languages under specific training strategies. Lastly, since the translationese affects model evaluation and performance, we give suggestions for future CLS data and model developments especially on low-resource languages.

**Contributions.** (1) To our knowledge, we are the first to investigate the influence of translationese on CLS. We confirm that the different translation methods in creating CLS datasets will lead to different degrees of translationese. (2) We conduct systematic experiments to show the effects of the translationese in source documents and target summaries, respectively. (3) Based on our findings, we discuss and give suggestions for future research.

## 2 Background

### 2.1 Translations in CLS Datasets

To provide a deeper understanding of the translations in CLS datasets, we comprehensively review previous datasets, and introduce the fountain of their documents and summaries, respectively.

Zhu et al. (2019) utilize a machine translation (MT) service to translate the summaries of two English monolingual summarization (MS) datasets (*i.e.*, CNN/Dailymail (Nallapati et al., 2016) and MSMO (Zhu et al., 2018)) to Chinese. The translated Chinese summaries together with the original English documents form En2ZhSum dataset. Later, Zh2EnSum (Zhu et al., 2019) and En2DeSum (Bai et al., 2021) are also constructed in this manner. The source documents of these CLS datasets are originally written in those languages (named *natural text*), while the target summaries are automatically translated from other languages (named *MT text*). Feng et al. (2022) utilize Google MT service[2] to translate both the documents and summaries of an English MS dataset (SAMSum (Gliwa et al., 2019)) into other five languages. The translated data together with the original data forms MSAMSum dataset. Thus, MSAMSum contains six source languages as well as six target languages, and only the English documents and summaries are *natural text*, while others are *MT text*. Since the translations provided by MT services might contain flaws, the above datasets further use round-trip translation strategy (§ 2.2) to filter out low-quality samples.

In addition to *MT text*, human-translated text (*HT text*) is also adopted in current CLS datasets. WikiLingua (Ladhak et al., 2020) collects document-summary pairs in 18 languages (including English) from WikiHow[3]. In this dataset, only the English documents/summaries are *natural text*, while all those in other languages are translated from the corresponding English versions by WikiHow's human writers (Ladhak et al., 2020). XSAMSum and XMediaSum (Wang et al., 2022a) are constructed through manually translating the summaries of SAMSum (Gliwa et al., 2019) and MediaSum (Zhu et al., 2021), respectively. Thus, the source documents and target summaries are *natural text* and

---

[2]https://cloud.google.com/translate
[3]https://www.wikihow.com/

| Dataset | Src Lang. | Text Type | Trg Lang. | Text Type |
|---|---|---|---|---|
| En2ZhSum | En | Natural | Zh | MT |
| Zh2EnSum | Zh | Natural | En | MT |
| En2DeSum | En | Natural | De | MT |
| MSAMSum | En Ar/Es/.../Zh | Natural MT | En Ar/Es/.../Zh | Natural MT |
| WikiLingua | En Ar/Cs/.../Zh | Natural HT | En Ar/Cs/.../Zh | Natural HT |
| XSAMSum | En | Natural | De/Zh | HT |
| XMediaSum | En | Natural | De/Zh | HT |
| XWikis | En/Fr/De/Cs | Natural & HT | En/Fr/De/Cs | Natural & HT |

Table 1: The summary of existing CLS datasets. "*Src Lang.*" and "*Trg Lang.*" denote the source and target languages in each dataset, respectively. "*Text Type*" represents the fountain of the source documents or target summaries ("*Natural*": natural text, "*MT*": Machine-translated text, "*HT*": human-translated text). Language nomenclature is based on ISO 639-1 codes.

*HT text*, respectively.

XWikis (Perez-Beltrachini and Lapata, 2021) collects document-summary pairs in 4 languages (*i.e.*, English, French, German and Czech) from Wikipedia. Each document-summary pair is extracted from a Wikipedia page. To align the parallel pages (which are relevant to the same topic but in different languages), Wikipedia provides Interlanguage links. When creating a new Wikipedia page, it is more convenient for Wikipedians to create by translating from its parallel pages (if has) than editing from scratch, leading to a large number of *HT text* in XWikis.[4] Thus, XWikis is formed with both *natural text* and *HT text*. Note that though the translations commonly appear in XWikis, we cannot distinguish which documents/summaries are *natural text* or *HT text*. This is because we are not provided with the translation relations among the parallel contents. For example, some documents in XWikis might be translated from their parallel documents, while others might be *natural text* serving as origins to create their parallel documents.

Table 1 summarizes the fountain of the source documents and target summaries in current datasets. We can conclude that when performing CLS from a source language to a different target language, translated text (*MT* and *HT text*) is extremely common in these datasets and appears more commonly in target summaries than source documents.

## 2.2 Round-Trip Translation

The round-trip translation (RTT) strategy is used to filter out low-quality CLS samples built by MT services. In detail, for a given text $t$ that needs to be translated, this strategy first translates $t$ into the target language $\hat{t}$, and then translates the result $\hat{t}$ back to the original language $t'$ based on MT services. Next, $\hat{t}$ is considered a high-quality translation if the ROUGE scores (Lin, 2004) between $t$ and $t'$ exceed a pre-defined threshold. Accordingly, the CLS samples will be discarded if the translations in them are not high-quality.

## 2.3 Translationese Metrics

To quantify translationese, we follow Toral (2019) and adopt automatic metrics from three aspects, *i.e.*, simplification, normalization and interference.

**Simplification.** Compared with natural text, translations tend to be simpler like using a lower number of unique words (Farrell, 2018) or content words (*i.e.*, nouns, verbs, adjectives and adverbs) (Scarpa, 2006). The following metrics are adopted:
- Type-Token Ratio (**TTR**) is used to evaluate lexical diversity (Templin, 1957) calculated by dividing the number of types (*i.e.*, unique tokens) by the total number of tokens in the text.
- Vocabulary Size (**VS**) calculates the total number of different words in the text.
- Lexical Density (**LD**) measures the information lies in the text by calculating the ratio between the number of its content words and its total number of words (Toral, 2019).

**Normalization.** The lexical choices in translated text tend to be normalized (Baker et al., 1993). We use entropy to measure this characteristic:
- Entropy of distinct $n$-grams (**Ent-n**) in the text.
- Entropy of content words (**Ent-cw**) in the text.

**Interference.** The structure of translated text tends to be similar to its source text (Gellerstam, 1996).
- Syntactic Variation (**SV**) is calculated by the *normalized tree edit distance* (Zhang and Shasha, 1989) between the constituency parse trees of the translated text and the source text.[5]
- Part-of-Speech Variation (**PSV**) is computed by the *normalized edit distance* between the part-of-speech sequences of the translated text and the source text.

It is worth noting that, ideally, the *fewer / lower-level* translationese in the translations, the *higher*

---

[4]https://en.wikipedia.org/wiki/Wikipedia:Translation

[5]We remove tokens from the constituency parse trees to let the metric focus on syntax rather than lexical.

all the above metrics will be.

## 3 Translationese in Target Summaries

In this section, we investigate how translationese affects CLS evaluation and training when it appears in the target summaries. For CLS datasets whose source documents are natural text while target summaries are HT text, we collect another summaries (in MT text) for them via Google MT. In this manner, one document will pair with two summaries (containing the same semantics, but one is HT text and the other is MT text). The translationese in these two types of summaries could be quantified. Subsequently, we can use the summaries in HT text and MT text as references, respectively, to train CLS models and analyze the influence of translationese on model performance.

### 3.1 Experimental Setup

**Datasets Selection.**

First, we should choose CLS datasets with source documents in natural text and target summaries in HT text. Under the consideration of the diversity of languages, scales and domains, we decide to choose XSAMSum (En⇒Zh) and WikiLingua (En⇒Ru/Ar/Cs).[6]

**Summaries Collection.** The original Chinese (Zh) summaries in XSAMSum, as well as the Russian (Ru), Arabic (Ar) and Czech (Cs) summaries in WikiLingua are HT text. Besides, XSAMSum and WikiLingua also provide the corresponding English summaries in natural text. Therefore, in addition to these original target summaries (in HT text), we can automatically translate the English summaries to the target languages to collect another summaries (in MT text) based on Google MT service.

RTT strategy (*c.f.*, Section 2.2) is further adopted to remove the low-quality translated summaries. As a result, the number of the translated summaries is less than that of original summaries. To ensure the comparability in subsequent experiments, we also discard the original summaries if the corresponding translated ones are removed. Lastly, the remaining original and translated summaries together with source documents form the final data we used.

Thanks to MSAMSum (Feng et al., 2022) which has already translated the English summaries of SAMSum to Chinese via Google MT service, thus,

---

[6]Since a CLS dataset might contain multiple source and target languages, we use "X⇒Y" to indicate the source language and target language are X and Y, respectively. Language nomenclature is based on ISO 639-1 codes.

| Statistics | XSAMSum | | WikiLingua | | | | | |
|---|---|---|---|---|---|---|---|---|
| Direction | En⇒Zh | | En⇒Ru | | En⇒Ar | | En⇒Cs | |
| Sum Type | HT | MT‡ | HT | MT† | HT | MT† | HT | MT† |
| Scale | 5929 | | 34273 | | 20751 | | 5686 | |
| TTR | **88.85** | 88.79 | **76.92** | 76.81 | 77.45 | **77.51** | **76.43** | 75.97 |
| VS | **12357** | 11207 | **44912** | 41782 | **20264** | 18538 | **16707** | 16063 |
| LD | **47.20** | 46.54 | **57.70** | 57.53 | **54.20** | 54.03 | 57.26 | **57.62** |
| Ent-1 | **3.932** | 3.917 | **4.94** | 4.88 | **4.75** | 4.67 | **5.01** | 4.98 |
| Ent-2 | **4.062** | 4.048 | **5.37** | 5.31 | **5.20** | 5.09 | **5.50** | 5.44 |
| Ent-cw | **2.960** | 2.928 | **4.02** | 3.96 | **3.99** | 3.96 | **3.95** | 3.91 |
| SV | **0.265** | 0.246 | - | - | - | - | - | - |
| PSV | **0.358** | 0.356 | **0.287** | 0.152 | **0.404** | 0.283 | **0.274** | 0.172 |

Table 2: Translationese statistics of target summaries. "*Direction*" indicates the source and target languages. "*Sum Type*" denotes the text type of target summaries (HT or MT text). "*Scale*" means the number of samples.

we directly utilize their released summaries[7] as the translated summaries of XSAMSum.

**Data Splitting.** After preprocessing, WikiLingua (En⇒Ru, En⇒Ar and En⇒Cs) contain 34,273, 20,751 and 5,686 samples, respectively. We split them into 30,273/2,000/2,000, 16,751/2,000/2,000 and 4,686/500/500 w.r.t training/validation/test sets. For XSAMSum, since the summaries in MT text are provided by Feng et al. (2022), we also follow their splitting, *i.e.*, 5307/302/320.

**Implementation Details.** Following recent CLS work (Feng et al., 2022; Wang et al., 2022a), we use mBART-50 (Tang et al., 2021) as the CLS model. The implementation details of model training and testing are given in Appendix B.

### 3.2 Translationese Analysis

We analyze the translationese in the target summaries of the preprocessed datasets. As shown in Table 2, the scores (measured by the metrics described in Section 2.3) in HT summaries are generally higher than those in MT summaries, indicating the HT summaries contain more diverse words and meaningful semantics, and their sentence structures are less influenced by the source text (*i.e.*, English summaries). Thus, the degree of translationese in HT summaries is less than that in MT summaries, which also verifies that different methods of collecting target-language summaries might lead to different degrees of translationese.

### 3.3 Translationese's Impact on Evaluation

For each dataset, we train two models with the same input documents but different target summaries. Specifically, one uses HT summaries as references (denoted as mBART-HT), while the other

---

uses MT summaries (denoted as mBART-MT).

Table 3 gives the experimental results in terms of ROUGE-1/2/L (R1/R2/R-L) (Lin, 2004) and BERTScore (B-S) (Zhang et al., 2020). Note that there are two ground-truth summaries (HT and MT) in the test sets. Thus, for model performance on each dataset, we report two results using HT and MT summaries as references to evaluate CLS models, respectively. It is apparent to find that when using MT summaries as references, mBART-MT performs better than mBART-HT, but when using HT summaries as references, mBART-MT works worse. This is because the model would perform better when the distribution of the training data and the test data are more consistent. Though straightforward, this finding indicates that if a CLS model achieves higher automatic scores on the test set whose summaries are MT text, it does not mean that the model could perform better in real applications where the translationese should be avoided.

To confirm the above point, we further conduct human evaluation on the output summaries of mBART-HT and mBART-MT. Specifically, we randomly select 100 samples from the test set of XSAMSum, and employ five graduate students as evaluators to score the generated summaries of mBART-HT and mBART-MT, and the ground-truth HT summaries in terms of informativeness, fluency and overall quality with a 3-point scale. During scoring, the evaluators are not provided with the source of each summary. More details about human evaluation are given in Appendix C. Table 4 shows the result of human evaluation. The Fleiss' Kappa scores (Fleiss, 1971) of informativeness, fluency and overall are 0.46, 0.37 and 0.52, respectively, indicating a good inter-agreement among our evaluators. mBART-HT outperforms mBART-MT in all metrics, and thus the human judgment is in line with the automatic metrics when adopting HT summaries (rather than MT summaries) as references. Based on this finding, we argue that when building CLS datasets, the translationese in target summaries of test sets should be carefully controlled.

### 3.4 Translationese's Impact on Training

Compared with HT summaries, when using MT summaries as references to train a CLS model, it is easier for the model to learn the mapping from the source documents to the simpler and more standardized summaries. In this manner, the generated sum-

| Ref.
Model | HT Summaries | MT Summaries |
|---|---|---|
| WikiLingua (En⇒Ru) | | |
| mBART-MT | 23.9 / 6.6 / 20.5 / 68.0 | **28.0 / 9.8 / 24.1 / 69.6** |
| mBART-HT | **24.6 / 8.0 / 21.5 / 68.2** | 23.9 / 6.8 / 20.7 / 67.8 |
| WikiLingua (En⇒Ar) | | |
| mBART-MT | 22.5 / 6.2 / 19.5 / 67.4 | **27.8 / 10.5 / 24.1 / 69.4** |
| mBART-HT | **23.6 / 7.6 / 20.9 / 67.8** | 23.0 / 6.9 / 20.2 / 67.3 |
| WikiLingua (En⇒Cs) | | |
| mBART-MT | 14.9 / 3.1 / 12.9 / 65.0 | **16.2 / 4.2 / 14.2 / 65.6** |
| mBART-HT | **16.5 / 4.1 / 14.6 / 65.2** | 15.4 / 3.8 / 13.6 / 64.9 |
| XSAMSum (En⇒Zh) | | |
| mBART-MT | 38.7 / 14.6 / 31.9 / 73.7 | **42.7 / 18.4 / 35.6 / 74.6** |
| mBART-HT | **39.1 / 14.9 / 32.2 / 74.2** | 40.2 / 15.6 / 33.1 / 74.2 |

Table 3: Experimental results (R1 / R2 / R-L / B-S) on WikiLingua and XSAMSum. "*Ref.*" indicates the references of each test set are HT summaries or MT summaries. mBART-HT and mBART-MT are trained with HT summaries and MT summaries, respectively.

| | Inform. | Fluency | Overall |
|---|---|---|---|
| mBART-MT | 2.06 | 1.53 | 1.77 |
| mBART-HT | 2.24 | 1.95 | 2.16 |
| Ground Truth | **2.42** | **2.81** | **2.73** |

Table 4: Human evaluation on the generated and ground-truth summaries (Inform.: Informativeness).

maries tend to have a good lexical overlap with the MT references since both the translationese texts contain normalized lexical usages. However, such summaries may not satisfy people in the real scene (*c.f.*, our human evaluation in Table 4). Thus, the translationese in target summaries during training has a negative impact on CLS model performance.

Furthermore, we find that mBART-HT has the following inconsistent phenomenon: the generated summaries of mBART-HT achieve a higher similarity with HT references than MT references on the WikiLingua (En⇒Ru, Ar and Cs) datasets (*e.g.*, 24.6 vs. 23.9, 23.6 vs. 23.0 and 16.5 vs. 15.4 R1, respectively), but are more similar to MT references on XSAMSum (*e.g.*, 40.2 vs. 39.1 R1). We conjecture this inconsistent performance is caused by the trade-off between the following factors: (i) mBART-HT is trained with the HT references rather than the MT references, and (ii) both the generated summaries and MT references are translationese texts containing normalized lexical usages. Factor (i) tends to steer the generated summaries closer to the HT references, while factor (ii) makes them closer to the MT references. When the CLS model has fully learned the mapping from the source documents to the HT summaries during training, factor (i) will dominate the generated sum-

| Statistics | WikiLingua | | | | | |
|---|---|---|---|---|---|---|
| Direction | Ar⇒En | | Ru⇒En | | Fr⇒En | |
| Doc Type | HT | MT | HT | MT | HT | MT |
| Scale | 25195 | | 36503 | | 60088 | |
| TTR | **60.45** | 59.45 | **60.01** | 58.48 | 46.15 | **46.55** |
| VS | **27360** | 25883 | **79306** | 77230 | **31285** | 30755 |
| LD | **50.75** | 49.73 | **51.95** | 50.73 | **44.00** | 43.74 |
| Ent-1 | **7.33** | 7.29 | **7.11** | 7.09 | **7.00** | 6.99 |
| Ent-2 | **8.41** | 8.41 | **8.31** | 8.23 | **8.58** | 8.55 |
| Ent-cw | **6.84** | 6.82 | **6.77** | 6.72 | **6.74** | 6.68 |
| PSV | **0.473** | 0.381 | **0.396** | 0.223 | **0.342** | 0.178 |

Table 5: Translationese statistics of source documents. "*Direction*" indicates the source and target languages. "*Doc Type*" denotes the text type of source documents.

| Source / Model | HT Documents | MT Documents |
|---|---|---|
| **WikiLingua (Ar⇒En)** | | |
| mBART-iMT | 32.6 / 10.9 / 26.8 / 71.0 | 34.7 / 12.8 / 28.7 / 71.8 |
| mBART-iHT | 32.9 / 11.5 / 27.3 / 71.2 | 33.5 / 12.1 / 27.7 / 71.3 |
| mBART-CL | 33.2 / 11.1 / 26.7 / 71.3 | ⟍ |
| mBART-TT | **33.9 / 11.8 / 27.8 / 71.6** | ⟍ |
| **WikiLingua (Ru⇒En)** | | |
| mBART-iMT | 32.7 / 11.2 / 27.0 / 71.0 | 35.0 / 13.3 / 29.1 / 71.7 |
| mBART-iHT | 32.9 / 11.6 / 27.3 / 70.9 | 33.5 / 12.2 / 27.8 / 71.1 |
| mBART-CL | 33.1 / 11.2 / 26.9 / 71.2 | ⟍ |
| mBART-TT | **33.8 / 12.1 / 28.0 / 71.5** | ⟍ |
| **WikiLingua (Fr⇒En)** | | |
| mBART-iMT | 33.6 / 11.9 / 28.2 / 71.3 | 36.6 / 14.5 / 30.8 / 72.6 |
| mBART-iHT | 34.8 / 13.0 / 29.2 / 71.7 | 35.0 / 13.2 / 29.3 / 71.9 |
| mBART-CL | 35.1 / 12.6 / 28.8 / 72.2 | ⟍ |
| mBART-TT | **35.8 / 13.2 / 29.7 / 72.5** | ⟍ |

Table 6: Experimental results (R1 / R2 / R-L / B-S) on WikiLingua (Ar/Ru/Fr⇒En). "*Source*" denotes using HT or MT documents as inputs to evaluate CLS models. mBART-iHT and mBART-iMT are trained with HT and MT documents, respectively. mBART-CL and mBART-TT are trained with both HT and MT documents via the curriculum learning and tagged training strategies, respectively. mBART-CL and mBART-TT are significantly better than mBART-iHT with t-test p < 0.05.

maries closer to the HT references, otherwise, the translationese in the generated summaries will lead them closer to the MT references. Therefore, the difficulty of CLS training data would lead to the inconsistent performance of mBART-HT. The verification of our conjecture is given in Appendix A.

## 4 Translationese in Source Documents

In this section, we explore how translationese affects CLS evaluation and training when it appears in the source documents. For CLS datasets whose source documents are HT text while target summaries are natural text, we collect another documents (in MT text) for them via Google MT service. In this way, one summary will correspond to two documents (containing the same semantics, but one is HT text and the other is MT text). Next, we can use the documents in HT text and MT text as source documents, respectively, to train CLS models and analyze the influence of translationese.

### 4.1 Experimental Setup

**Datasets Selection.** The CLS datasets used in this section should contain source documents in HT text and target summaries in natural text. Here, we choose WikiLingua (Ar/Ru/Fr⇒En).

**Documents Collection.** To collect MT documents, we translate the original English documents of WikiLingua to Arabic (Ar), Russian (Ru) and French (Fr) via Google MT service. Similar to Section 3.1, RTT strategy is also adopted to control the quality.

**Data Splitting.** After preprocessing, WikiLingua (Ar/Ru/Fr⇒En) contain 25,195/36,503/60,088 samples. We split them into 21,195/2,000/2,000, 30,503/3,000/3,000 and 54,088/3,000/3,000 w.r.t training/validation/test sets, respectively.

### 4.2 Translationese Analysis

We analyze the translationese in the preprocessed documents. Table 5 shows that most scores of HT documents are higher than those of MT documents, indicating a lower degree of translationese in HT documents. Thus, the different methods to collect the source documents might also result in different degrees of translationese.

### 4.3 Translationese's Impact on Evaluation

For each direction in the WikiLingua dataset, we train two mBART models with the same output summaries but different input documents. In detail, one uses HT documents as inputs (denoted as mBART-iHT), while the other uses MT documents (denoted as mBART-iMT).

Table 6 lists the experimental results in terms of ROUGE-1/2/L (R-1/2/L) and BERTScore (B-S). Note that there are two types of input documents (HT and MT) in the test sets. For each model, we report two results using HT and MT documents as inputs to generate summaries, respectively. Compared with using HT documents as inputs, both mBART-iHT and mBART-iMT achieve higher automatic scores when using MT documents as inputs. For example, mBART-iHT achieves 32.7 and 33.7 R1, using HT documents and MT documents as inputs in WikiLingua (Ru⇒En), respectively. The counterparts of mBART-iMT are 32.4

and 34.8 R1. In addition to the above automatic evaluation, we conduct human evaluation on these four types (mBART-iHT/iMT with HT/MT documents as inputs) of the generated summaries. In detail, we randomly select 100 samples from the test set of WikiLingua (Ar⇒En). Five graduate students are asked as evaluators to assess the generated summaries in a similar way to Section 3.3. For evaluators, the parallel documents in their mother tongue are also displayed to facilitate evaluation. As shown in Table 7, though using MT documents leads to better results in terms of automatic metrics, human evaluators prefer the summaries generated using HT documents as inputs. Thus, automatic metrics like ROUGE and BERTScore cannot capture human preferences if input documents are machine translated. Besides, translationese in source documents should also be controlled in the test sets.

## 4.4 Translationese's Impact on Training

When using HT documents as inputs, mBART-iHT outperforms mBART-iMT in both automatic and human evaluation (Table 6 and Table 7). Thus, the translationese in source documents during training has a negative impact on CLS model performance. However, different from the translationese in summaries, the translationese in documents do not affect the training objectives. Consequently, we wonder if it is possible to train a CLS model with both MT and HT documents and further improve the model performance. In this manner, MT documents are also utilized to build CLS models, benefiting the research on low-resource languages. We attempt the following strategies: (1) **mBART-CL** heuristically adopts a curriculum learning (Bengio et al., 2009) strategy to train a mBART model from ⟨MT document, summary⟩ samples to ⟨HT document, summary⟩ samples in each training epoch. (2) **mBART-TT** adopts the tagged training strategy (Caswell et al., 2019; Marie et al., 2020) to train a mBART model. The strategy has been studied in machine translation to improve the MT performance on low-resource source languages. In detail, the source inputs with high-level translationese (*i.e.*, MT documents in our scenario) are prepended with a special token [TT]. For other inputs with low-level translationese (*i.e.*, HT documents), they remain unchanged. Therefore, the special token explicitly tells the model these two types of inputs.

As shown in Table 6, both mBART-CL and mBART-TT outperform mBART-iHT in all three di-

|  | Inform. | Fluency | Overall |
|---|---|---|---|
| mBART-iMT (input MT doc.) | 1.62 | 1.84 | 1.74 |
| mBART-iMT (input HT doc.) | 1.89 | 2.05 | 2.07 |
| mBART-iHT (input MT doc.) | 2.17 | 2.29 | 2.11 |
| mBART-iHT (input HT doc.) | 2.39 | 2.43 | 2.23 |
| Ground Truth | **2.78** | **2.84** | **2.88** |

Table 7: Human evaluation on summaries (Inform.: Informativeness, doc.: document).

rections (according to the conclusion of our human evaluation, we only use HT documents as inputs to evaluate mBART-CL and mBART-TT). Besides, mBART-TT outperforms mBART-CL, confirming the superiority of tagged training in CLS. To give a deeper analysis of the usefulness of MT documents, we use a part of (10%, 30%, 50% and 70%, respectively) HT documents (paired with summaries) and all MT documents (paired with summaries) to jointly train mBART-TT model. Besides, we use the same part of HT documents to train mBART-iHT model for comparisons. Table 8 gives the experimental results. With the help of MT documents, mBART-TT only uses 50% of HT documents to achieve competitive results with mBART-iHT. Note that compared with HT documents, MT documents are much easier to obtain, thus the strategy is friendly to low-resource source languages.

## 5 Discussion and Suggestions

Based on the above investigation and findings, we conclude this work by presenting concrete suggestions to both the dataset and model developments.
**Controlling translationese in test sets.** As we discussed in Section 3.3 and Section 4.3, the translationese in source documents or target summaries would lead to the inconsistency between automatic evaluation and human judgment. In addition, one should avoid directly adopting machine-translated documents or summaries in the test sets of CLS datasets. To make the machine-translated documents or summaries suitable for evaluating model performance, some post-processing strategies should be conducted to reduce translationese. Prior work (Zhu et al., 2019) adopts post-editing strategy to manually correct the machine-translated summaries in their test sets. Though post-editing increases productivity and decreases errors compared to translation from scratch (Green et al., 2013), Toral (2019) finds that post-editing machine translation also has special stylistic which is different from native usage, *i.e.*, post-editese. Thus, the post-

| | HT | MT | WikiLingua (Ar⇒En) | WikiLingua (Ru⇒En) | WikiLingua (Fr⇒En) |
|---|---|---|---|---|---|
| mBART-iHT | ✔ (10%) | ✗ | 26.9 / 7.4 / 21.8 / 68.3 | 27.9 / 7.6 / 22.5 / 68.5 | 29.7 / 9.1 / 24.2 / 69.7 |
| | ✔ (30%) | ✗ | 29.6 / 9.2 / 24.2 / 69.7 | 29.9 / 9.2 / 24.5 / 69.7 | 31.7 / 10.6 / 26.1 / 70.4 |
| | ✔ (50%) | ✗ | 31.4 / 10.3 / 25.8 / 70.5 | 31.3 / 10.3 / 25.7 / 70.3 | 33.6 / 11.9 / 27.7 / 71.4 |
| | ✔ (70%) | ✗ | 32.2 / 11.0 / 26.5 / 70.9 | 31.9 / 10.9 / 26.5 / 70.7 | 33.9 / 12.2 / 28.2 / 71.5 |
| | ✔ (100%) | ✗ | 32.9 / 11.6 / 27.3 / 71.2 | 32.9 / 11.6 / 27.3 / 70.9 | 34.8 / 13.0 / 29.2 / 71.7 |
| mBART-TT | ✔ (10%) | ✔ (100%) | 31.8 / 10.6 / 26.2 / 70.4 | 32.9 / 11.2 / 27.1 / 71.0 | 34.4 / 12.3 / 28.7 / 71.8 |
| | ✔ (30%) | ✔ (100%) | 32.7 / 11.1 / 26.9 / 71.3 | 32.9 / 11.3 / 27.2 / 71.0 | 34.7 / 12.6 / 28.7 / 71.8 |
| | ✔ (50%) | ✔ (100%) | 33.4 / 11.5 / 27.3 / 71.3 | 33.0 / 11.5 / 27.2 / 70.9 | 34.6 / 12.5 / 28.8 / 71.9 |
| | ✔ (70%) | ✔ (100%) | 33.8 / **11.9** / 27.7 / 71.5 | 33.7 / 12.0 / 27.8 / 71.4 | 35.2 / 12.9 / 29.2 / 72.2 |
| | ✔ (100%) | ✔ (100%) | **33.9** / 11.8 / **27.8** / **71.6** | **33.8** / **12.1** / **28.0** / **71.5** | **35.8** / **13.2** / **29.7** / **72.5** |

Table 8: Experimental results (R1 / R2 / R-L / B-S). "*HT*" and "*MT*" indicate the percentages of ⟨HT document, summary⟩ and ⟨MT document, summary⟩ pairs used to train each CLS model, respectively. The **bold** and underline denote the best and the second scores, respectively.

editing strategy cannot reduce translationese. Future studies can explore other strategies to control translationese in CLS test sets.

**Building mixed-quality or semi-supervised CLS datasets.** Since high-quality CLS pairs are difficult to collect (especially for low-resource languages), it is costly to ensure the quality of all samples in a large-scale CLS dataset. Future work could collect mix-quality CLS datasets that involve both high-quality and low-quality samples. In this manner, the collected datasets could encourage model development in more directions (*e.g.*, the curriculum learning and tagged training strategies we discussed in Section 4.4) while controlling the cost. In addition, grounding the truth that monolingual summarization samples are much easier than CLS samples to collect under the same resources (Hasan et al., 2021b,a), semi-supervised datasets, which involve CLS samples and in-domain monolingual summarization samples, are also a good choice.

**Designing translationese-aware CLS models.** It is inevitable to face translationese when training CLS models. The degree of translationese might be different in the document or summary of each training sample when faced with one of the following scenarios: (1) training multi-domain CLS models based on multiple CLS datasets; (2) training CLS models in low-resource languages based on mixed-quality datasets. In this situation, it is necessary to build translationese-aware CLS models. The tagged training strategy (Caswell et al., 2019) is a simple solution which only considers two-granularity translationese in source documents. It is more general to model the three-granularity translationese (*i.e.*, natural text, HT text and MT text) in documents as well as summaries. Future work could attempt to (1) explicitly model the different degrees of translationese (such as prepending special tokens), or (2) implicitly let the CLS model be aware of different degrees of translationese (*e.g.*, designing auxiliary tasks in multi-task learning).

## 6 Related Work

**Cross-Lingual Summarization.** Cross-lingual summarization (CLS) aims to summarize source-language documents into a different target language. Due to data scarcity, early work typically focuses on pipeline methods (Leuski et al., 2003; Wan et al., 2010; Wan, 2011; Yao et al., 2015), *i.e.*, translation and then summarization or summarization and then translation. Recently, many large-scale CLS datasets are proposed one after another. According to an extensive survey on CLS (Wang et al., 2022b), they can be divided into synthetic datasets and multi-lingual website datasets. Synthetic datasets (Zhu et al., 2019; Bai et al., 2021; Feng et al., 2022; Wang et al., 2022a) are constructed by translating monolingual summarization (MS) datasets. Multi-lingual website datasets (Ladhak et al., 2020; Perez-Beltrachini and Lapata, 2021) are collected from online resources. Based on these large-scale datasets, many researchers explore various ways to build CLS systems, including multi-task learning strategies (Cao et al., 2020; Liang et al., 2022), knowledge distillation methods (Nguyen and Luu, 2022; Liang et al., 2023a), resource-enhanced frameworks (Zhu et al., 2020) and pre-training techniques (Xu et al., 2020; Wang et al., 2022a, 2023b; Liang et al., 2023b). More recently, Wang et al. (2023a) explore zero-shot CLS by prompting large language models. Different from them, we are the first to investigate the influence of translationese on CLS.

**Translationese.** Translated texts are known to

have special features which refer to "translationese" (Gellerstam, 1986). The phenomenon of translationese has been widely studied in machine translation (MT). Some researchers explore the influence of translationese on MT evaluation (Lembersky et al., 2012; Zhang and Toral, 2019; Graham et al., 2020; Edunov et al., 2020). To control the effect of translationese on MT models, tagged training (Caswell et al., 2019; Marie et al., 2020) is proposed to explicitly tell MT models if the given data is translated texts. Besides, Artetxe et al. (2020) and Yu et al. (2022) mitigate the effect of translationese in cross-lingual transfer learning.

## 7 Conclusion

In this paper, we investigate the influence of translationese on CLS. We design systematic experiments to investigate how translationese affects CLS model evaluation and performance when translationese appears in source documents or target summaries. Based on our findings, we also give suggestions for future dataset and model developments.

## Ethical Considerations

In this paper, we use mBART-50 (Tang et al., 2021) as the CLS model in experiments. During fine-tuning, the adopted CLS samples mainly come from WikiLingua (Ladhak et al., 2020), XSAMSum (Wang et al., 2022a) and MSAMSum (Feng et al., 2022). Some CLS samples might contain translationese and flawed translations (provided by Google Translation). Therefore, the trained models might involve the same biases and toxic behaviors exhibited by these datasets and Google Translation.

## Limitations

While we show the influence of translationese on CLS, there are some limitations worth considering in future work: (1) We do not analyze the effects of translationese when the translations appear in both source documents and target summaries; (2) Our experiments cover English, Chinese, Russian, Arabic, Czech and French, and future work could extend our method to more languages and give more comprehensive analyses w.r.t different language families.

## Acknowledgements

This work is supported by the National Natural Science Foundation of China (No.62072323, U21A20488, 62206056), Shanghai Science and Technology Innovation Action Plan (No. 22511104700), and Key Projects of Industrial Foresight and Key Core Technology Research and Development in Suzhou (SYC2022009).

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

| Dataset | Scale | Coverage |
|---|---|---|
| WikiLingua (En⇒Ru) | 34,273 | 57.58 |
| WikiLingua (En⇒Ar) | 20,751 | 58.68 |
| WikiLingua (En⇒Cs) | 5,686 | 57.54 |
| XSAMSum | 5,929 | 37.99 |

Table 9: The scales and coverage of CLS data.

| Ref. / Test Set | HT summaries | MT summaries |
|---|---|---|
| Hard Subset | 16.9 / 4.9 / 15.6 | 17.6 / 6.6 / 16.6 |
| Simple Subset | 16.5 / 4.1 / 14.5 | 15.2 / 3.5 / 13.4 |

Table 10: Results of mBART-HT (R1 / R2 / R-L) on the hard and simple test subsets of WikiLingua (En⇒Cs).

4154–4164, Brussels, Belgium. Association for Computational Linguistics.

Junnan Zhu, Qian Wang, Yining Wang, Yu Zhou, Jiajun Zhang, Shaonan Wang, and Chengqing Zong. 2019. NCLS: Neural cross-lingual summarization. In *Proceedings of the 2019 Conference on Empirical Methods in Natural Language Processing and the 9th International Joint Conference on Natural Language Processing (EMNLP-IJCNLP)*, pages 3054–3064, Hong Kong, China. Association for Computational Linguistics.

Junnan Zhu, Yu Zhou, Jiajun Zhang, and Chengqing Zong. 2020. Attend, translate and summarize: An efficient method for neural cross-lingual summarization. In *Proceedings of the 58th Annual Meeting of the Association for Computational Linguistics*, pages 1309–1321, Online. Association for Computational Linguistics.

# A The Inconsistent Performance of mBART-HT

According to our conjecture, XSAMSum (En⇒Zh) should be more difficult than WikiLingua (En⇒Ru/Ar/Cs) for CLS models to perform. Consequently, factor (i) dominates in WikiLingua, while factor (ii) dominates in XSAMSum, leading to the inconsistent performance. To convince that, we illustrate the difficulty of each CLS dataset from the following aspects: (1) **Scale** calculates the number of CLS samples in each dataset. Generally, the more samples used to train a CLS model, the easier it is for the model to learn CLS. (2) **Coverage** measures the overlap rate between documents and summaries, which is defined as the average proportion of the copied bigram in summaries for each dataset.[8] The higher coverage of a dataset, the less

---

[8] Since the documents and summaries in CLS datasets are in different languages, the coverage is calculated based on

search space for models to learn the mapping from source documents to target summaries. As shown in Table 9, XSAMSum is more difficult than other datasets based on the overall consideration from both aspects. To provide a deeper explanation of our conjecture, we further evenly split the test set of WikiLingua (En⇒Cs) into hard and simple subsets according to the coverage of each sample. The coverage of samples in the hard subset is less that in the simple subset. As shown in Table 10, the generated summaries of mBART-HT are closer to HT references on the simple subset, but more similar to MT references on the hard subset, demonstrating that the difficulty of CLS training data would lead to the inconsistent performance of mBART-HT.

## B Implementation Details

The implementation of mBART-50 (Tang et al., 2021) (610M parameters) used in our experiments is provided by the Huggingface Transformers.[9] We fine-tune the model on NVIDIA Tesla V100 GPUs (32G) and set the learning rate to 5e-6, the warmup steps to 500, the epochs to 10, and the batch size is 4. The maximum number of tokens for input sequences is 1024. In the test process, beam size is set to 5. All experimental results listed in this paper are the average of 3 runs.

To calculate ROUGE scores, we employ the *multilingual ROUGE* toolkit[10] that considers segmentation and stemming algorithms for various languages. To calculate BERTScore, we use the *bertscore* toolkit[11].

## C Human Evaluation

We tell our evaluators a brief guideline about three metrics: (1) Informativeness measures how informative the summary is. (2) Fluency measures how fluent, and grammatical the summary is. Is a summary well-written and grammatically correct? (3) Overall measures the overall quality of each generated summary. It can be judged under the consideration of informativeness, fluency, relevance, consistency and so on. Then, all evaluators are required to give each summary a score selected from "1", "2" and "3" for each metric. When making the

judgments, all summaries of a given document are provided for our evaluators simultaneously to let them make comparisons among different models (note that all evaluators do not know every summary is generated by which model, and the appearance order of summaries is shuffled). We do not provide detailed breakdown for each score in each metric due to the following reasons: (1) We encourage each evaluator to follow their actual feelings to make judgments since everyone in the real applications might have different criteria (for each metric). For example, someone might be sensitive to fluency while others might for informativeness. Thus, we want to make our human evaluation more in line with this real-world scenario. (2) It is hard and even unrealistic to construct a perfect quantitative human evaluation principle. (3) This evaluation method is commonly used in machine translation evaluation (Callison-Burch et al., 2007; Denkowski and Lavie, 2010), cross-lingual summarization evaluation (Zhu et al., 2020; Cao et al., 2020; Liang et al., 2022) and other tasks (Li et al., 2023b,a).

---

the source-language summaries (*i.e.*, English summaries in WikiLingua and XSAMSum) and source documents.

[9]https://huggingface.co/facebook/mbart-large-50-many-to-many-mmt

[10]https://github.com/csebuetnlp/xl-sum/tree/master/multilingual_rouge_scoring

[11]https://github.com/Tiiiger/bert_score