# OpenReview forum: "Understanding Translationese in Cross-Lingual Summarization"
_EMNLP/2023/Conference — EMNLP 2023 Findings_

### Official Review · Reviewer_AJcm · 2023-08-03

**Soundness:** 3

**Excitement:**

4: Strong: This paper deepens the understanding of some phenomenon or lowers the barriers to an existing research direction.

**Paper Topic And Main Contributions:**

In the authors' words, the aim of the work is to confirm that
different approaches of constructing cross-lingual summarization (CLS)
datasets will lead to different degrees of translationese. Then they
study how such translationese affects CLS model evaluation and
performance when it appears in source documents or target summaries.

**Questions For The Authors:**

The authors' responses seem conditioned by the size limitations of the
text, both in my case and in that of the rest of the reviewers.

Certainly some of the modifications required imply a significant
investment of time and work, but it is no less true that if this is
the case, it is because it refers to aspects that should have been
considered before writing the text. More precisely, we have that:

A. Please, justify the authors' position in relation to the reviewer's
item A in the previous section "Reasons to reject".

Regarding the authors' responses, it is not clear whether or not some type of
comparison with other operating models will be included in the
reviewed paper, given that the authors limit themselves to stating
that:

"We would like to add this discussion in Limitation section"

when I have no doubt that it is an issue that must be addressed in one
way or another in an eventual final review, if only to warn of the
possibility of biases associated with the use of a certain type of
modeling, given that this would condition the entire experimental part
of the work.

B. Please, justify the authors' position in relation to the reviewer's
item B in the previous section "Reasons to reject".

Regarding the authors' answers at this point, indeed most existing translationese-relevant
studies typically focus on MT, which would have precisely facilitated
a comparison like the one we have just commented on. In any case, we
are talking about a decision of the authors who have adequately
justified.

**Reasons To Accept:**

The impact of translationese in the domain of NLP is a critical
issue. Here it is important to take into account its growing
dependence on ML techniques, a reality associated with the need for
training corpus in different fields of knowledge and languages. This
has led to the application of TM tools to its generation and,
consequently, the generalization of all kinds of phenomena related to
the concept of "translationese". In turn, given the popularization of
NLP-based tools, this could even affect language on a longer term. The
recent irruption of GPT-like models delves into the importance of
studying this type of problems, particularly in relation to
summarization.

In this context, the work is interesting. Regardless of the fact that
the authors' conclusions coincide with those intuitively expected, the
truth is that the evidence corroborates them and seems to dispel any
possible doubts.

**Reasons To Reject:**

A. I wonder if the fact of considering only m-BART-based MT models for testing
does not introduce some kind of bias in the results.

B. I also wonder if the best way to address the impact of
translationese in the field of cross-lingual technologies would not be
by focusing our attention on the basic MT technologies... and not on
each of their possible practical applications (summatization, ...). In
this regard, it should be remembered that there are already works
published in this line as, for example:

Eva Vanmassenhove, Dimitar Shterionov, and Matthew
Gwilliam. 2021. Machine Translationese: Effects of Algorithmic Bias on
Linguistic Complexity in Machine Translation. In Proceedings of the
16th Conference of the European Chapter of the Association for
Computational Linguistics: Main Volume, pages 2203–2213,
Online. Association for Computational Linguistics.

Sicheng Yu, Qianru Sun, Hao Zhang, and Jing
Jiang. 2022. Translate-Train Embracing Translationese Artifacts. In
Proceedings of the 60th Annual Meeting of the Association for
Computational Linguistics (Volume 2: Short Papers), pages 362–370,
Dublin, Ireland. Association for Computational Linguistics.

the latter of which is one of the references included in this work.

**Reproducibility:**

5: Could easily reproduce the results.

**Reviewer Confidence:**

4: Quite sure. I tried to check the important points carefully. It's unlikely, though conceivable, that I missed something that should affect my ratings.

**Typos Grammar Style And Presentation Improvements:**

Is such a display of bibliographic references really necessary ? Their
presence is overwhelming and makes it difficult to understand the text
(see for example Lines 040 or 058). Such display does not at all favor
the reading and comprehension of the text.

Please, avoid redirects in the text, particularly in the Introduction
(see Line 063, 083, 084). Instead, include a "road-map" of the paper
at the end of Section 1, including a brief look of all the other
Sections.

Most of the last paragraph (Contributions) of Section 1 retake the
same contents included in the previous one as well in the Abstract. Is
this really necessary.

---

> ### Author Rebuttal · Authors · 2023-08-28
>
> Dear Reviewer,
>
> Thank you for your comments concerning our manuscript entitled “Understanding Translationese in Cross-Lingual Summarization”. We thank you for the very helpful suggestions and we provide below with our answers to your comments. We have followed closely the suggestions, and made clarifications and revisions accordingly. We hope the newly provided content could help to further strengthen our work.
>
> **Comment 1**: I wonder if the fact of considering only m-BART-based MT models for testing does not introduce some kind of bias in the results.
> **Response to comment 1**: There is no doubt that conducting experiments on more multi-lingual pre-trained models (like mT5, BLOOM or LLaMA) would give more comprehensive analyses. Since the human evaluation is labor-intensive and time-consuming, we only conduct experiments on mBART-based model, and it might involve bias of mBART itself. We would like to add this discussion in Limitation section.
> Besides, we want to emphasize that the mBART model is a commonly used end-to-end baseline in the cross-lingual summarization research field [1-4]. This is why we choose to conduct experiments on this model. We sincerely hope the current settings could be accepted.
>
> [1] WikiLingua: A New Benchmark Dataset for Cross-Lingual Abstractive Summarization (EMNLP 2020 Findings)
> [2] Models and Datasets for Cross-Lingual Summarisation (EMNLP 2021)
> [3] A Variational Hierarchical Model for Neural Cross-Lingual Summarization (ACL 2022)
> [4] Towards Unifying Multi-Lingual and Cross-Lingual Summarization (ACL 2023)
>
> **Comment 2**: I also wonder if the best way to address the impact of translationese in the field of cross-lingual technologies would not be by focusing our attention on the basic MT technologies... and not on each of their possible practical applications (summatization, ...). In this regard, it should be remembered that there are already works published in this line as, for example Machine Translationese: Effects of Algorithmic Bias on Linguistic Complexity in Machine Translation (EACL 2021).
> **Response to comment 2**: To the best of our knowledge, most existing translationese-relevant studies typically focus on machine translation (MT) which is also a practical application. In this paper, we want to provide analytical research from another lens, i.e., cross-lingual summarization (CLS). Different from MT, CLS is a more complex scene that requires the model to perform both translation and summarization. Though translationese is already studied in MT, the current CLS research field typically neglects this phenomenon, and the traditional evaluation (ROUGE or BERTScore) might not reflect human feelings towards CLS systems in real scenes. Thus, we want to openly discuss these potential risks and they can cause problems. As for your recommended reference, we will cite it in the related work in the revised version.
>
> **Comment 3**: About Typos Grammar Style And Presentation Improvements
> **Response to comment 3**: Thank you for pointing them out! We will sincerely consider your constructive suggestions in the revised version.

---

### Official Review · Reviewer_WQsK · 2023-08-04

**Soundness:** 3

**Ethical Concerns:**

Yes

**Excitement:**

4: Strong: This paper deepens the understanding of some phenomenon or lowers the barriers to an existing research direction.

**Justification For Ethical Concerns:**

Details on human evaluation are missing, it is not stated how much the evaluators were paid or whether the study design was reviewed by IRB.

**Paper Topic And Main Contributions:**

In this paper the authors explore the effect of translationese on the cross-lingual summarization (CLS). Since in the cross-lingual summarization task the data is often obtained via translation (often MT) this leads to a question of whether the translationese, likely introduced during this process, affects the task itself (CLS)

The authors investigate how translationese, when present in source documents or target summaries, affects the modeling and evaluation of CLS. The author explore this problem through a series of experiments and give suggestions on how one may deal with this issue in the future. These include controlling translationese in test sets, developing translationese-aware models, and building mixed-quality CLS datasets.

The authors also include a partial human evaluation, though details are missing.

**Questions For The Authors:**

Question A: How were the human evaluators selected? Were they compensated for this task?

Question B: What is the 3-point scale? What instructions were given to the annotators?

**Reasons To Accept:**

- The problem addressed in this study is very timely and likely to be an issue in the current systems;

- The experiment choice is motivated by the RQs.

**Reasons To Reject:**

- I am not sure about the human evaluation employed, as there are no sufficient details in the paper raising ethical and methodological questions. For instance, I believe that the ratings should not be displayed as means as the scale employed is likely ordinal (or at least a distribution of scores should be added. This further raises a question about the IAA analysis, as the reported Fleiss Kappa should be employed to categorical data. In other words, the IAA is done for categorical data, the means are reported as if the data was interval/numerical, while most likely the data is ordinal, though it is impossible to say as the details are not reported (or somehow I have missed that).

**Reproducibility:**

4: Could mostly reproduce the results, but there may be some variation because of sample variance or minor variations in their interpretation of the protocol or method.

**Reviewer Confidence:**

3: Pretty sure, but there's a chance I missed something. Although I have a good feel for this area in general, I did not carefully check the paper's details, e.g., the math, experimental design, or novelty.

**Typos Grammar Style And Presentation Improvements:**

Apart from Fleiss Kappa, it would be also good to report the percentage agreement. If the data is not categorical, Krippendorff's alpha should be employed. Preferably CI should be reported.

---

> ### Author Rebuttal · Authors · 2023-08-28
>
> Dear Reviewer,
>
> Thank you for your comments concerning our manuscript entitled “Understanding Translationese in Cross-Lingual Summarization”. We thank you for the very helpful suggestions and we provide below with our answers to your comments. We have followed closely the suggestions, and made clarifications and revisions accordingly. We hope the newly provided content could help to further strengthen our work.
>
> **Comment 1**: The authors also include a partial human evaluation, though details are missing.
> **Response to comment 1**: Thank you for pointing it out!  We tell our evaluators a brief guideline about the three human evaluation metrics we used: (1) Informativeness measures how informative the summary is. (2) Fluency measures how fluent, and grammatical the summary is. Is a summary well-written and grammatically correct? (3) Overall measures the overall quality of each generated summary. It can be judged under the consideration of informativeness, fluency, relevance, consistency and so on.
> Then, all evaluators are required to give each summary a score selected from “1”, “2” and “3” for each metric. When making the judgments, all summaries of a given document are provided for our evaluators simultaneously to let them make comparisons among different models (note that all evaluators do not know each summary is generated by which model, and the appearance order of summaries is shuffled).
> We will add these details of our human evaluation in the revised version.
>
> **Comment 2**: At least a distribution of human evaluation scores should be added.
> **Response to comment 2**: We provide the distributions of our human evaluation scores in the following tables:
>
> System |Metric|proportion of “1”|proportion of “2”|proportion of “3”|average score
> -|-|-|-|-|-
> mBART-mt|inform.|36.0|22.0|42.0|2.06
> mBART-ht|inform.|19.2|37.6|43.2|2.24
> Ground Truth|inform.|7.0|44.0|49.0|2.42
> mBART-mt|fluency|62.6|21.8|15.6|1.53
> mBART-ht|fluency|35.2|34.6|30.2|1.95
> Ground Truth|fluency|4.4|10.2|85.4|2.81
> mBART-mt|overall|48.2|26.6|25.2|1.77
> mBART-ht|overall|27.2|29.6|43.2|2.16
> Ground Truth|overall|10.0|7.0|83.0|2.73
>
> (The distributions of human evaluation scores in Section 3)
>
> System |Metric|proportion of “1”|proportion of “2”|proportion of “3”|average score
> -|-|-|-|-|-
> mBART-iMT (input MT doc.)|inform.|56.0|26.0|18.0|1.62
> mBART-iMT (input HT doc.)|inform.|42.0|27.0|31.0|1.89
> mBART-iHT (input MT doc.)|inform.|28.6|25.8|45.6|2.17
> mBART-iHT (input HT doc.)|inform.|20.0|21.0|59.0|2.39
> Ground Truth|inform.|8.2|5.6|86.2|2.78
> mBART-iMT (input MT doc.)|fluency|43.0|30.0|27.0|1.84
> mBART-iMT (input HT doc.)|fluency|32.4|30.2|37.4|2.05
> mBART-iHT (input MT doc.)|fluency|23.4|24.2|52.4|2.29
> mBART-iHT (input HT doc.)|fluency|19.0|19.0|62.0|2.43
> Ground Truth|fluency|6.2|3.6|90.2|2.84
> mBART-iMT (input MT doc.)|overall|51.2|23.6|25.2|1.74
> mBART-iMT (input HT doc.)|overall|30.8|31.4|37.8|2.07
> mBART-iHT (input MT doc.)|overall|29.4|30.2|40.4|2.11
> mBART-iHT (input HT doc.)|overall|29.8|17.4|52.8|2.23
> Ground Truth|overall|3.8|4.4|91.8|2.88
>
> (The distributions of human evaluation scores in Section 4)
> Please also note that the human evaluation samples are randomly selected from the test set of CLS data, and there is no order among them.
>
> **Comment 3**: The reported Fleiss Kappa should be employed to categorical data.
> **Response to comment 3**: We provide the Fleiss' Kappa scores with respect to different subsets of summaries as following tables:
>
> Fleiss' Kappa |Informativeness|Fluency|Overall
> -|-|-|-
> 100 summaries generated by mBART-MT|0.41|0.32|0.48
> 100 summaries generated by mBART-HT|0.41|0.33|0.45
> 100 ground truth summaries|0.48|0.36|0.62
> all 300 summaries|0.46|0.37|0.52
>
> (The Fleiss' kappa scores of human study in Section 3 w.r.t different subsets of summaries)
>
> Fleiss' Kappa |Informativeness|Fluency|Overall
> -|-|-|-
> 100 summaries generated by mBART-iMT (input MT docs)|0.48|0.40|0.44
> 100 summaries generated by mBART-iMT (input HT docs)|0.51|0.52|0.60
> 100 summaries generated by mBART-iHT (input MT docs)|0.50|0.48|0.54
> 100 summaries generated by mBART-iHT (input HT docs)|0.45|0.41|0.44
> 100 ground truth summaries|0.55|0.50|0.60
> all 500 summaries|0.56|0.52|0.58
>
> (The Fleiss' kappa scores of human study in Section 4 w.r.t different subsets of summaries)
>
> **Comment 4**: Question A: How were the human evaluators selected? Were they compensated for this task?
> **Response to comment 4**: (1) We employ five Chinese graduate students who are fluent in English as our evaluators to conduct human evaluation. All evaluators have extensive experience in data annotation (participating many human studies in our lab's previous work), and a small number (about 20%) of their evaluation results are checked by a data expert to ensure the quality of human evaluation.  (2) During the human evaluation, the salary for annotating each summary is determined by the average time of annotation and local labor compensation standard. We would like to add this explanation in Ethical Considerations.
>
> **Comment 5**: Question B: What is the 3-point scale? What instructions were given to the annotators?
> **Response to comment 5**: Please refer to response to comment 1.
>
> **Comment 6**: Apart from Fleiss Kappa, it would be also good to report the percentage agreement. If the data is not categorical, Krippendorff's alpha should be employed. Preferably CI should be reported.
> **Comment 6.1**: It would be also good to report the percentage agreement.
> **Response to comment 6.1**: Thank you for pointing it out! Since the human evaluation scores are annotated under the comparisons among different models, we provide the percentage agreements of model comparisons as follows:
>
> Percentage Agreements |mBART-MT|mBART-HT|Ground Truth
> -|-|-|-
> mBART-MT|-|58%|70%
> mBART-HT|58%|-|65%
> Ground Truth|70%|65%|-
>
> (The percentage agreements of human evaluation in Section 3)
>
> Percentage Agreements |mBART-iMT (input MT doc)|mBART-iMT (input HT doc) |mBART-iHT (input MT doc)|mBART-iHT (input HT doc)|Ground Truth
> -|-|-|-|-|-
> mBART-iMT (input MT doc)|-|53%|57%|62%|74%
> mBART-iMT (input HT doc)|53%|-|53%|61%|70%
> mBART-iHT (input MT doc)|57%|53%|-|60%|73%
> mBART-iHT (input HT doc)|62%|61%|60%|-|69%
> Ground Truth|74%|70%|73%|69%|-
>
> (The percentage agreements of human evaluation in Section 4)
>
> Each percentage in the above tables means what percentage of the test samples achieves consistent results among all five evaluators when comparing the corresponding two models (including ground truth). The consistent results mean all five evaluators (1) give the first model a better score, (2) give the second model a better score, or (3) give the same score for both models.
>
> **Comment 6.2**: If the data is not categorical, Krippendorff's alpha should be employed. Preferably CI should be reported.
> **Response to comment 6.2**: Our human evaluation score is categorical. Please also refer to the response to comment 1.

---

### Official Review · Reviewer_pmRA · 2023-08-09

**Typos Grammar Style And Presentation Improvements:** 1.  latex errors
**Soundness:** 3

**Excitement:**

3: Ambivalent: It has merits (e.g., it reports state-of-the-art results, the idea is nice), but there are key weaknesses (e.g., it describes incremental work), and it can significantly benefit from another round of revision. However, I won't object to accepting it if my co-reviewers champion it.

**Paper Topic And Main Contributions:**

Paper is a comprehensive look at the impact of translationese in cross-lingual summarization.  authors motivate the work based on the shortage of cross-lingual summarization resources.  After introducing some metrics to estimate translationese, they look at their impact when present in the source text, target summaries and also their impact on summarization models.  Results indicate their presence and motivates some suggestions about containing their presence with automatic and manual efforts.

**Questions For The Authors:**

1. where are details of the wide range of linguistic analysis to compute translationese metrics (parse, pos, entropy, etc).  how do factor varying levels of quality for such analysis in different languages?

2. section 3.3 is quite confusing:  here the direction is en2x and one would expect the summaries to be in the x languages and annotation is conducted in those 4-5 languages?  also any explanation for the fair/weak agreement for fluency and also the fact that overal agreement is higher than the individual factors?

3. Re: Table 8, the MT data is always kept at 100% in the ablation study.  do you think using a cleaner subset of the MT data (using your translationese metrics) can make the impact of that data stronger?

**Reasons To Accept:**

1. Interesting topic (cross-language summarization).

2. Fairly comprehensive look at the impact of translationese in source, target, eval and models.  The comprehensive work just requires some substantial polishing and restructuring.

**Reasons To Reject:**

1. Paper has a poor writing and it is difficult to follow.  Moreover it is not organized well (long sentences, latex mistakes, many accronyms, long repetitions of table results in the text body). Even some basic concepts like translationese are not defined well.  There's not a single example in the entire paper.

2. poor replicability of the work.  so much details are missing in computing various translationese metrics for different languages and cls models.

3. The conclusion and macro level impact of the work does not seem significant.

**Reproducibility:**

3: Could reproduce the results with some difficulty. The settings of parameters are underspecified or subjectively determined; the training/evaluation data are not widely available.

**Reviewer Confidence:**

3: Pretty sure, but there's a chance I missed something. Although I have a good feel for this area in general, I did not carefully check the paper's details, e.g., the math, experimental design, or novelty.

---

> ### Author Rebuttal · Authors · 2023-08-28
>
> Dear Reviewer,
>
> Thank you for your comments concerning our manuscript entitled “Understanding Translationese in Cross-Lingual Summarization”. We thank you for the very helpful suggestions and we provide below with our answers to your comments. We have followed closely the suggestions, and made clarifications and revisions accordingly. We hope the newly provided content could help to further strengthen our work.
>
> **Comment 1**: Paper has a poor writing and it is difficult to follow. Moreover, it is not organized well (long sentences, latex mistakes, many acronyms, long repetitions of table results in the text body). Even some basic concepts like translationese are not defined well. There's not a single example in the entire paper.
>
> **Comment 1.1**: Paper has a poor writing and it is difficult to follow. Moreover it is not organized well (long sentences, latex mistakes, many acronyms, long repetitions of table results in the text body).
> **Comment 1.2**: Typos Grammar Style And Presentation Improvements: (1) latex errors: line 057 (long referencing at the middle of sentence); (2) Table 9 & 10 in page 8.
> **Response to comments 1.1 & 1.2**: Please kindly let us know if something in our manuscript is difficult to follow, and we would appreciate it. (1) For latex mistakes in line 57, we promise to revise the long referencing in the middle. (2) For acronyms, we explain the full meaning of each acronym the first time it occurs, and we hope this could be accepted. (3) As for long repetitions of table results in the text body. Since our work is an analytical study, we have tried our best to analyze the phenomenon in experiments, and it is inevitable to repeat some results. (4) About Tables 9 & 10 in page 8: both Table 9 and table 10 belong to Appendix, and they are in page 11 currently.
>
> **Comment 1.3**: Even some basic concepts like translationese are not defined well.
> **Response to comment 1.3**: We explain translationese in Lines 64-78, and the definition follows the main-stream definition like the first paragraph in [1]. We hope this definition can be accepted.
>
> [1] Translationese as a Language in “Multilingual” (ACL 2020)
>
> **Comment 1.4**: There's not a single example in the entire paper
> **Response to comment 1.4**: Currently, we do not provide any cases of translationese since (1) it is recognized that the translated or model-generated text might involve translationese. (2) space limitation. If this work could be accepted, we promise to add a case in additional page of content.
>
> [2] Post-editese: an Exacerbated Translationese (MTSummit 2019)
>
> **Comment 2**: poor replicability of the work. so much details are missing in computing various translationese metrics for different languages and cls models.
> **Response to comment 2**: Actually, our translationese metrics typically follow previous translationese work [2] (as mentioned in **line 221**). We also would like to share the full version of the calculating scripts once publication. Here, we provide the key implementation details of these metrics as follows:
> (1)Type-Token Ratio (TTR) and vocabulary size (VS): To calculate TTR and VS, we need to tokenize the input sentence, here we use mBART50's tokenizer with the following Python code:
> ```python
> from transformers import MBart50TokenizerFast
> tokenizer = MBart50TokenizerFast.from_pretrained("facebook/mbart-large-50")
>
> input_sentence = "..."
> tokenized_sentence = tokenizer.tokenize(input_sentence)
> ```
> (2) Lexical Density (LD):  To calculate LD, we need the POS tags of input sentences in different languages. We use [stanza](https://stanfordnlp.github.io/stanza/) to obtain them ([stanza supports many languages](https://stanfordnlp.github.io/stanza/available_models.html)) via the following Python code:
> ```python
> import stanza
> lang2stanzacode = {
>     'arabic': 'ar',
>     'chinese': 'zh',
>     'russian': 'ru',
>     'czech': 'cs',
>     'french': 'fr',
> } # these languages are used in our experiments
>
> nlp = stanza.Pipeline(lang2stanzacode[lang], processors='tokenize,pos', use_gpu=True)
>
> content_words = 0 # number of content words
> total_words = 0 # number of all words
>
> input_sentence = "..."
> stanza_doc = nlp(input_sentence)
> for sent in stanza_doc.sentences:
>     for word in sent.words:
>         if word.pos in ['ADJ','ADV','VERB','NOUN']:
>             content_words += 1
>         total_words += 1
>
> lexical_density = content_words / float(total_words)
> ```
> (3) Entropy of distinct n-grams (Ent-n) and Entropy of content words (Ent-cw): These two metrics are also calculated based on tokens (using mBART50's tokenizer)
> (4) Syntactic Variation (SV): The constituency parse trees are calculated via Stanza:
> ```python
> import stanza
>
> nlp = stanza.Pipeline("zh", processors='tokenize,pos,constituency', use_gpu=True) # calculating constituency parse trees for Chinese
>
> Chinese_sentence = "..."
> stanza_doc = nlp(Chinese_sentence)
>
> constituency_parse_tree = zh_sum_xsamsum.sentences[0].constituency
> ```
> Please also note that the stanza's constituency algorithm [does not support Arabic, Russian, Czech and French](https://stanfordnlp.github.io/stanza/constituency.html), thus we only calculate SV for Chinese text (see **Table 2** in our paper)
> (5) Part-of-Speech Variation (PSV): The POS tags used to calculate PSV are obtained via the same method as LD.
>
> **Comment 3**: Where are details of the wide range of linguistic analysis to compute translationese metrics (parse, pos, entropy, etc). how do factor varying levels of quality for such analysis in different languages?
> **Comment 3.1**: Where are details of the wide range of linguistic analysis to compute translationese metrics (parse, pos, entropy, etc).
> **Response to comment 3.1**: Our translationese metrics typically follow previous translationese work [2] (as mentioned in **line 221**). The effectiveness of these metrics to measure translationese has already been verified in previous work. Therefore, we do not give details of linguistic analysis for them, and please do not worry about these metrics.
>
> [2] Post-editese: an Exacerbated Translationese (MTSummit 2019)
>
> **Comment 3.2**: how do factor varying levels of quality for such analysis in different languages?
> **Response to comment 3.2**: We think the translationese evaluation is another interesting research topic, but it goes beyond this work. In this work, we use existing metrics to analyze the influence of translationese in cross-lingual summarization systems, rather than propose new translationese metrics. We sincerely hope the current analysis could be accepted.
>
> **Comment 4**: Section 3.3 is quite confusing: here the direction is en2x and one would expect the summaries to be in the x languages and annotation is conducted in those 4-5 languages? also any explanation for the fair/weak agreement for fluency and also the fact that overal agreement is higher than the individual factors?
> **Comment 4.1**: here the direction is en2x and one would expect the summaries to be in the x languages and annotation is conducted in those 4-5 languages?
> **Response to comment 4.1**: Yes, for a given English document, there are annotated summaries in multiple languages.
>
> **Comment 4.2**: Any explanation for the fair/weak agreement for fluency
> **Response to comment 4.2**: We think the Fleiss' kappa scores we achieved (0.46, 0.37 and 0.52 in Section 3) are decent compared with previous cross-lingual summarization work. For example, the Fleiss' kappa scores in [3] are 0.20, 0.22 and 0.37 (c.f., Table 4 in [3]), and the scores in [4] are 0.26, 0.37 and 0.43 (c.f., Section 6.3 in [4]). Therefore, please do not worry about the degree of evlautors' agreement. We also additionally provide the percentage agreements of human evaluation to ensure the soundness of our human evaluation, please refer to **the response to reviewer WQsK's comment 6.1** if you are interested.
>
> [3] Cross-Lingual Abstractive Summarization with Limited Parallel Resources (ACL 2021)
> [4] ClidSum: A Benchmark Dataset for Cross-Lingual Dialogue Summarization (EMNLP 2022)
>
> **Comment 4.3**: Any explanation for the fact that overal agreement is higher than the individual factors.
> **Response to comment 4.3**: We do not provide detailed guidelines for each score in each metric (i.e., informativess, fluency or overall), and the scores are annotated via comparing outputs among different models. Therefore, the results of different metrics are not comparable. In addition, if you are interested in the process of our human evaluation, please refer to **the response to reviewer pBBk's comment 1.2**.
>
> **Comment 5**: Table 8, the MT data is always kept at 100% in the ablation study. do you think using a cleaner subset of the MT data (using your translationese metrics) can make the impact of that data stronger?
> **Response to comment 5**: These translationese metrics are used to measure text distributions rather than a single sentence or document. Therefore, using them to select a cleaner MT subset might have potential risks and need to be further verified, which is beyond the scope of this work. If we can select a cleaner MT subset, we think there might be a trade-off between quality and quantity, thus it needs experiments.

---

### Official Review · Reviewer_pBBk · 2023-08-10

**Paper Topic And Main Contributions:** 1. This paper explores the impact of …
**Soundness:** 4

**Excitement:**

4: Strong: This paper deepens the understanding of some phenomenon or lowers the barriers to an existing research direction.

**Reasons To Accept:**

1. This paper is the first attempt to analyze the impact of translationese in cross-lingual summarization tasks.
2. The authors provided a detailed analysis of the impact of translationese concerning source documents and Target summaries for various languages and datasets.
3. The authors highlight findings, such as the potential promise in constructing mixed-quality or semi-supervised cross-lingual summarization datasets, which could be a valuable direction for further research.
4. This paper emphasizes how important it is to use human-translated documents and summaries to create strong cross-lingual summarization models.
5. The analysis process used to identify the impact of translationese allows other groups to reproduce for different language families.

**Reasons To Reject:**

1. In section 3.3, the human evaluation part isn't clear.

      a. Were the evaluators native speakers? What was their expertise level?

      b. What guidelines were given for human evaluation? The specifics of the 3-point scale breakdown for metrics like Informativeness,
          Fluency, and Overall are missing.

     c.  On line 363, what does "overall quality" mean?

     d. The authors mentioned selected 100 samples from the XSAMSUM test set randomly. How many of these belonged to mBART-HT
           and mBART-MT? However, Table 4 indicates separate human evaluation for these subsets.

     e. If human evaluation was done separately, I would expect additional Fleiss' kappa scores for Informativeness, Fluency, and Overall.

     f. On line 369, mentioned that  "good agreement between evaluators." But according to Fleiss' kappa score interpretation
           (https://en.wikipedia.org/wiki/Fleiss%27_kappa), the scores show fair to moderate agreement. It's important to explain the reasons
          for the lower agreement scores.


      g. Opting for human evaluation on two different datasets would be preferable.

2. The above raised issue apply to section 4.3 also, where Fleiss' kappa scores and metric ratings are missing.

3. As stated in Appendix B, all reported experimental scores are averages of 3 runs. However, the corresponding standard deviation scores are absent.

4. Regarding lines 022-023, there's no evidence or empirical study provided on how translationese might negatively affect CLS model performance in real-world applications.

**Reproducibility:**

5: Could easily reproduce the results.

**Reviewer Confidence:**

4: Quite sure. I tried to check the important points carefully. It's unlikely, though conceivable, that I missed something that should affect my ratings.

**Typos Grammar Style And Presentation Improvements:**

Minor issue:
1. line183 --> check the spacing

---

> ### Author Rebuttal · Authors · 2023-08-28
>
> Dear Reviewer,
>
> Thank you for your comments concerning our manuscript entitled “Understanding Translationese in Cross-Lingual Summarization”. We thank you for the very helpful suggestions and we provide below with our answers to your comments. We have followed closely the suggestions, and made clarifications and revisions accordingly. We hope the newly provided content could help to further strengthen our work.
>
> **Comment 1**: In section 3.3, the human evaluation part isn't clear.
> **Comment 1.1**: Were the evaluators native speakers? What was their expertise level?
> **Response to comment 1.1**: We employ five **Chinese** graduate students who are fluent in **English** as our evaluators to conduct human evaluation on both Section 3 (Table 4) and Section 4 (Table 7). (1) The human evaluation in Section 3 is conducted with English documents and Chinese summaries (i.e., XSAMSum En->Zh), thus, our evaluators can deal with both involved languages. (2) The human evaluation in Section 4 is conducted with Arabic documents and English summaries (i.e., Wikilingua Ar->En). To facilitate the evaluation on Arabic documents, the Chinese parallel documents are also shown to our evaluators (as we mentioned in **Lines 480-481**). Note that, the Chinese parallel documents are provided by the original WikiLingua dataset and contain the same semantics as the Arabic documents. In this way, our evaluators can focus on the generated English summaries and give their judgments about the summary quality.
>
> **Comment 1.2**: What guidelines were given for human evaluation? The specifics of the 3-point scale breakdown for metrics like Informativeness, Fluency, and Overall are missing.
> **Response to comment 1.2**: Actually, we do not make the 3-point scale breakdown, and we only tell our evaluators a brief guideline about three metrics: (1) Informativeness measures how informative the summary is. (2) Fluency measures how fluent, and grammatical the summary is. Is a summary well-written and grammatically correct? (3) Overall measures the overall quality of each generated summary. It can be judged under the consideration of informativeness, fluency, relevance, consistency and so on.
> Then, all evaluators are required to give each summary a score selected from “1”, “2” and “3” for each metric. When making the judgments, all summaries of a given document are provided for our evaluators simultaneously to let them to make comparsions among different models (note that all evaluators do not know every summary is generated by which model, and the appearance order of summaries is shuffled).
> We do not provide detailed guidelines/breakdown for each score in each metric due to the following reasons: (1) We encourage each evaluator to follow their actual feelings to make judgments since everyone in the real applications might have different criteria (for each metric). For example, someone might be sensitive to fluency while others might for informativeness. Thus, we want to make our human evaluation more in line with this real-world scenario.  (2) It is hard and even unrealistic to construct a perfect quantitative human evaluation principle.  (3) This evaluation method is commonly used in machine translation evaluation [1,2], and cross-lingual summarization evaluation [3-5].
>
> [1] (Meta-) Evaluation of Machine Translation (WMT 2007)
> [2] Choosing the Right Evaluation for Machine Translation: an Examination of Annotator and Automatic Metric Performance on Human Judgment Tasks  (AMTA 2010)
> [3] Attend, Translate and Summarize: An Efficient Method for Neural Cross-Lingual Summarization (ACL 2020)
> [4] Jointly Learning to Align and Summarize for Neural Cross-Lingual Summarization (ACL 2020)
> [5] A Variational Hierarchical Model for Neural Cross-Lingual Summarization (ACL 2022)
>
> **Comment 1.3**: On line 363, what does "overall quality" mean?
> **Response to comment 1.3**: Please refer to response to comment 1.2
>
> **Comment 1.4**: The authors mentioned selected 100 samples from the XSAMSUM test set randomly. How many of these belonged to mBART-HT and mBART-MT? However, Table 4 indicates separate human evaluation for these subsets. If the human evaluation was done separately, I would expect additional Fleiss' kappa scores for Informativeness, Fluency, and Overall.
> **Response to comment 1.4**: (1) Each sample from the test set means an input document and the corresponding ground truth summary. We randomly select 100 samples from the test set, and use mBART-HT and mBART-MT to generate their 100 summaries, respectively. Thus, we have 100 summaries generated by mBART-HT and other 100 summaries generated by mBART-MT. Besides, there are also 100 ground truth summaries. (2) The Fleiss' kappa scores of informativeness, fluency, and overall on all 300 summaries are 0.46, 0.37 and 0.52 (as we mentioned in **Lines 367-368**).  The additional Fleiss' kappa scores of each subset are listed in the following table:
>
> Fleiss' Kappa |Informativeness|Fluency|Overall
> -|-|-|-
> 100 summaries generated by mBART-MT|0.41|0.32|0.48
> 100 summaries generated by mBART-HT|0.41|0.33|0.45
> 100 ground truth summaries|0.48|0.36|0.62
> all 300 summaries|0.46|0.37|0.52
>
> **Comment 1.5**: On line 369, mentioned that "good agreement between evaluators." But according to Fleiss' kappa score interpretation, the scores show fair to moderate agreement. It's important to explain the reasons for the lower agreement scores.
> **Response to comment 1.5**: We think the Fleiss' kappa scores we achieved (0.46, 0.37 and 0.52 in Section 3) are decent compared with previous cross-lingual summarization work. For example, the Fleiss' kappa scores in [6] are 0.20, 0.22 and 0.37 (c.f., Table 4 in [6]), and the scores in [7] are 0.26, 0.37 and 0.43 (c.f., Section 6.3 in [7]). Therefore, please do not worry about the degree of evaluators' agreement.  We also additionally provide the percentage agreements of human evaluation to ensure the soundness of our human evaluation, please refer to **the response to reviewer WQsK's comment 6.1** if you are interested.
>
> [6] Cross-Lingual Abstractive Summarization with Limited Parallel Resources (ACL 2021)
> [7] ClidSum: A Benchmark Dataset for Cross-Lingual Dialogue Summarization (EMNLP 2022)
>
> **Comment 1.6**: Opting for human evaluation on two different datasets would be preferable.
> **Response to comment 1.6**: Thanks for this constructive suggestion! Due to the space limitation, we did human evaluation on XSAMSum (En->Zh) in Section 3, and on WikiLingua (Ar->En) in Section 4. We think current results have already verified our claims and showed the influence of translationese in cross-lingual summarization. We sincerely hope the current settings can be accepted, and we also would like to take this suggestion into account in the future.
>
> **Comment 2**: The above raised issue apply to section 4.3 also, where Fleiss' kappa scores and metric ratings are missing.
> **Response to comment 2**: Thank you for pointing out this! The human evaluation metrics used in Section 4 are the same as those in Section 3. The Fleiss' kappa scores of human study in Section 4 are listed in the following table:
>
> Fleiss' Kappa |Informativeness|Fluency|Overall
> -|-|-|-
> 100 summaries generated by mBART-iMT (input MT docs)|0.48|0.40|0.44
> 100 summaries generated by mBART-iMT (input HT docs)|0.51|0.52|0.60
> 100 summaries generated by mBART-iHT (input MT docs)|0.50|0.48|0.54
> 100 summaries generated by mBART-iHT (input HT docs)|0.45|0.41|0.44
> 100 ground truth summaries|0.55|0.50|0.60
> all 500 summaries|0.56|0.52|0.58
>
> **Comment 3**: As stated in Appendix B, all reported experimental scores are averages of 3 runs. However, the corresponding standard deviation scores are absent.
> **Response to comment 3**: To ensure the soundness of this study, we conduct each experiment three times and report the average results. When we conduct the experiments multiple times, we find the results in terms of ROUGE scores and BERTScore are almost the same, and the standard deviation scores are typically less than 0.1. Therefore, we do not report the standard deviation scores. Besides, reporting standard deviation scores is not a common practice in the cross-lingual summarization research field.
>
> **Comment 4**: Regarding lines 022-023, there's no evidence or empirical study provided on how translationese might negatively affect CLS model performance in real-world applications.
> **Response to comment 4**: We are sorry for this misunderstanding. The “real-world applications” here means “how people really feel about the model-generated cross-lingual summaries when using CLS systems”. We want to use “real-world applications” to emphasize our study goes beyond automatic evaluations which may not match what humans really feel. To this end, we use human evaluation in both Section 3 and Section 4 to show the influence of translationese in CLS systems.

---

### Meta-Review · Area_Chair_w7XX · 2023-09-16

**Recommendation:** 4

**Metareview:**

The paper presents a study of the impact of “translationese” on Cross Lingual Summarization (CLS), a problem relevant to the CLS area. An evaluation based on human annotators is presented, including Inter Annotator Agreement values. Some interesting conclusions are drawn from the work.

Some unresolved questions remain:
- Is Fleiss' Kappa appropriate for calculating IAA when using an ordinal scale (R3)?
- Does it make sense to study the effect of translationese in a particular application, such as summarization, or does it make more sense to study it in Machine Translation methods in general (R4)?
- It is important to better describe the annotation task that was performed (R1 and R2).

---

### Decision · Program_Chairs · 2023-10-07

**Decision:**

Accept-Findings

**Comment:**

The paper presents a study of the impact of “translationese” on Cross Lingual Summarization (CLS), a problem relevant to the CLS area. An evaluation based on human annotators is presented, including Inter Annotator Agreement values. Some interesting conclusions are drawn from the work.

Some unresolved questions remain:
- Is Fleiss' Kappa appropriate for calculating IAA when using an ordinal scale (R3)?
- Does it make sense to study the effect of translationese in a particular application, such as summarization, or does it make more sense to study it in Machine Translation methods in general (R4)?
- It is important to better describe the annotation task that was performed (R1 and R2).